# Regulation of Transcription Factor NF-κB in Its Natural Habitat: The Nucleus

**DOI:** 10.3390/cells10040753

**Published:** 2021-03-29

**Authors:** Susanne Bacher, Johanna Meier-Soelch, Michael Kracht, M. Lienhard Schmitz

**Affiliations:** 1Institute of Biochemistry, Justus-Liebig-University, D-35392 Giessen, Germany; susanne.bacher@biochemie.med.uni-giessen.de; 2Rudolf-Buchheim-Institute of Pharmacology, Justus-Liebig-University, D-35392 Giessen, Germany; johanna.soelch@pharma.med.uni-giessen.de (J.M.-S.); Michael.Kracht@pharma.med.uni-giessen.de (M.K.)

**Keywords:** transcription factor, NF-κB, chromatin, epigenetics, genome organization, stem cells, signaling

## Abstract

Activation of the transcription factor NF-κB elicits an individually tailored transcriptional response in order to meet the particular requirements of specific cell types, tissues, or organs. Control of the induction kinetics, amplitude, and termination of gene expression involves multiple layers of NF-κB regulation in the nucleus. Here we discuss some recent advances in our understanding of the mutual relations between NF-κB and chromatin regulators also in the context of different levels of genome organization. Changes in the 3D folding of the genome, as they occur during senescence or in cancer cells, can causally contribute to sustained increases in NF-κB activity. We also highlight the participation of NF-κB in the formation of hierarchically organized super enhancers, which enable the coordinated expression of co-regulated sets of NF-κB target genes. The identification of mechanisms allowing the specific regulation of NF-κB target gene clusters could potentially enable targeted therapeutic interventions, allowing selective interference with subsets of the NF-κB response without a complete inactivation of this key signaling system.

## 1. Cytosolic and Nuclear NF-κB Regulation

The nuclear factor kappa B (NF-κB) signaling module is a complex and highly interconnected molecular network, with important functions in, probably, all nucleated cells [1]. NF-κB can be perceived as an evolutionary conserved system that enables cells to cope with many types of stress, ranging from inflammatory stimuli to oxidative, nutritional, and even physical stressors [2]. In addition, NF-κB plays an important role in differentiation and developmental processes, owing to its ability to counteract cell death, improve survival, and trigger the expression of secreted differentiation factors [3]. In support of its broad relevance, the dysregulation of NF-κB can be (co)causative for a number of ailments including chronic inflammation and the acquired chemotherapy resistance of cancer cells [4]. Canonical NF-κB activation involves two types of regulatory steps, as schematically shown in Figure 1A:(I)A mostly cytosolic response where receptor-derived signals are converted into a cascade of enzymatic reactions. Activation of NF-κB can proceed via three main pathways, namely the canonical, noncanonical, and atypical NF-κB activation cascades, as described in detail in a number of excellent reviews [1,2]. A central and common hallmark of all pathways is the proteolytic destruction of inhibitor of NF-κB (IκB) proteins, which serve to retain the DNA-binding dimers in the cytosol. Proteasome-mediated removal of IκB enables the nuclear accumulation of NF-κB dimers, where they can bind to DNA and reprogram gene expression [5]. These key cytosolic events feed into NF-κB activation events, but also into NF-κB independent processes. This diversity can even be integrated within the same enzyme with dual functions, such as the IκB kinase (IKK) complex, which leads to the phosphorylation and degradation of IκB proteins, but also phosphorylates numerous further substrate proteins with relevance in other pathways, including regulation of mRNA stability [6] or mammalian target of rapamycin (mTOR) signaling [7].(II)A mostly nuclear response, where the five different NF-κB DNA-binding subunits p65 (also known as RELA), REL (also known as c-Rel), RELB, p50, and p52 form various DNA-binding dimers to reprogram gene expression [8]. RELA, REL, and RELB harbor transactivation domains in their C-termini that serve to induce transcription in a target gene-specific fashion [9]. Transactivation domains are absent from the family members NF-κB2 (also known as p100) and NF-κB1 (also known as p105), and these precursor proteins are processed either during translation or through phosphorylation-induced partial proteolysis to yield p52 and p50, respectively [10]. As schematically displayed in Figure 1A, the nuclear response is highly regulated at multiple levels, to allow precise control of all parameters relevant to NF-κB-driven gene expression as a prerequisite for an appropriate biological response. This involves the physical interaction of NF-κB subunits with many nuclear proteins, as visualized by an interaction network of the subunits with further proteins, using the STRING database (https://string-db.org/, accessed on 4 March 2021) (Figure 1B). However, these interactions have been largely determined by classical biochemical methods, which favor the detection of stable interactions. We still lack a comprehensive understanding of the full repertoire of the nuclear NF-κB interactome, including all possible unstable, low affinity, and substoichiometric protein–protein interactions.

NF-κB-mediated signaling and transcription can occur in several modes, as schematically depicted in the left part of Figure 1C. Stimulus-triggerd NF-κB activation is typically concentration-dependent and transient, as is the downstream gene response. Sub-physiological levels of NF-κB triggers will lead to chronically lowered gene responses and to immune-compromised phenotypes, as many NF-κB target genes regulate innate and adaptive immune functions [11]. The continuous presence of sub-maximal concentrations of NF-κB stimuli will promote chronic low grade activation, which is often observed in the tumor microenvironment or in chronic diseases associated with “smoldering” or low-grade inflammation (e.g., atherosclerosis) [12]. Progressively increasing NF-κB activation can occur slowly during aging or senescence, which are both characterized by metabolic changes and a chronic increase in secreted inflammatory cytokines, resulting in the so-called senescence-associated secretory phenotype (SEAP) [13,14,15]. While a pulsed stimulus results in a single peak of strongly activated gene expression, persistent or recurrent exposure to high concentrations of NF-κB triggers an oscillatory NF-κB response, as it is also reflected by periodically elevated levels of TNFα in rheumatoid arthritis or IL-1β in chronic gout arthritis [16]. This type of NF-κB activation can also result from the induction of distinct feedback mechanisms at various time-delayed time points [17,18]. A major unresolved question in the NF-κB field is, by which principles these functional modes of NF-κB activation are mechanistically linked to the complex organization of the 3D genome, as schematically displayed in Figure 1C, right part.

## 2. NF-κB Regulation in Stem Cells

The various NF-κB activation pathways also occur in stem cells and cellular components of the nervous system, along with the different shades of differential NF-κB activity. Here, we will not focus on the cytosolic NF-κB pathways occurring in neuronal cells and stem cells, but rather discuss recent insights into the function of NF-κB in the nucleus. Hematopoietic stem and progenitor cells (HSPCs) display a tumor necrosis factor receptor (TNFR)-associated factor 6 (TRAF6)-dependent low level of basal NF-κB activity in the absence of inflammatory signals [19]. These hematopoietic stem cells experience slightly increased NF-κB activity during aging, a process that is accompanied by elevated inflammation and decreased stem cell function [20]. However, a further increase of NF-κB activity in HSPCs, induced by overexpression of a constitutively active IKKß kinase, leads to deregulation of transcription factor networks and concomitant loss of quiescence, as well as bone marrow suppression [21,22]. While HSPCs do not tolerate maximal NF-κB activation, other stem cell types can fully activate this transcription factor, either in pathophysiological situations in response to damage or inflammatory insults, or in physiological settings, where NF-κB-regulated cytokines regulate multiple fates of hematopoietic stem cells, including survival, quiescence, self-renewal, differentiation, and mobilization from their niche [23].

Compared to specialized immune cells such as monocytes/macrophages or cultured cancer cell lines, considerably less information on nuclear NF-κB regulation is available for neuronal and stem cells. Stem cells and differentiating cells can be distinguished by several features from terminally differentiated cells, including factors that control lineage specification and development such as PU.1 [24]. This transcription factor binds to enhancers in cells of the myeloid and B lineages and indirectly causes histone H3 lysine 4 monomethylation (H3K4me1) and nucleosome repositioning. These events provide an open chromatin environment that allows inducible DNA-binding of NF-κB in response to its activation [25]. The analysis of human embryonic stem (ES) cells and lineages derived thereof revealed extensive chromatin changes during lineage specification that affected more than one third of the genome and were characterized by massive changes in chromatin organization [26]. These major architectural changes also affect 3D folding and organization of the genome and thereby the chromatin accessibility of NF-κB. Although these rather fundamental differences between stem cells and differentiated cells have been observed, the available evidence also suggests that most general principles of the multi-level nuclear NF-κB response are independent of the cell type [27].

## 3. Structural Organization of the Genome and Its Connection to the NF-κB System

Due to the spatial constraints of the nucleus, all nuclear molecules and their functions have to be highly organized at various levels, as schematically depicted in Figure 1C, right part. The chromosomes localize into so-called chromosome territories after mitosis [28]. These chromosome territories can be further partitioned into distinct compartments, which are enriched for transcriptionally active euchromatin (A compartments) or transcriptionally repressed heterochromatin (B compartments), respectively [29]. An ever increasing portfolio of methods, including chromosome conformation capture (3C) assays and microscopic techniques, has revealed further levels of chromosomal sub-organization, in particular the topologically associated domains (TADs). These genomic regions typically show multiple long-range contacts, mainly between enhancers and gene promoters within the same TAD [30,31]. The TAD boundaries are enriched with binding sites for the DNA-binding factor CCCTC-binding factor (CTCF) [32]. CTCF and its associated factors serve as insulators of the TAD region and thereby promote the co-regulation of functionally related genes, such as cytokine gene clusters, within a particular TAD [33]. High-resolution mapping of the genome allowed the identification of the sub-TADs that are associated with enrichment of specific chromatin marks [34]. Finally, the smallest packing units of the genome are the nucleosomes, consisting of about 150 base pairs of DNA sequence wrapped around a core of histone proteins.

## 4. Regulation of NF-κB in the Nucleus

NF-κB activating stimuli affect all different levels of nuclear organization, as discussed in this section.

### 4.1. Global Chromatin Organization

Stimulation of cells with proinflammatory cytokines such as TNFα and interleukin 1 (IL-1ß) leads to refolding of large chromatin compartments [35,36,37]. In human umbilical vein endothelial (HUVEC) cells, TNFα-induced changes in genome organisation lead to the co-association of many TNFα-responsive genes, congregating into discrete “NF-κB factories” [38]. These focal areas of high NF-κB activity were indirectly visualized by RNA fluorescence in situ hybridization, which detected spatial clustering of nascent RNAs [38]. TNFα also induces changes in the higher-order chromatin conformation in the TNF/lymphotoxin locus on chromosome 6, as revealed by 3C assays [39]. Moreover, lipopolysaccharide (LPS)-stimulated monocytes show stimulus-induced long-range DNA interactions that move the genes at the *IL1*/*IL36*/*IL37* gene cluster on chromosome 2 into close proximity [40]. IL-1α stimulation of cells led to a reshuffling of higher-order chromatin interactions on chromosome 4 (in *cis*), as judged by native chromosome conformation capture (i4C) interactome profiles [36]. This study and a further report also revealed abundant IL-1-regulated changes in chromatin accessibility throughout the genome at the level of nucleosomes, as determined by ATAC-seq (assay for transposase-accessible chromatin using sequencing) [36,41].

The driving forces for these global 3D rewiring processes might employ loop extrusion, self-organization, or transcription [42]. Loop extrusion was suggested to proceed via progressive extrusion of chromatin loops through the cohesin ring at CTCF-bound chromatin regions until it reaches convergent CTCF-bound sites located on the same or neighbouring TADs [43]. Self-organization largely proceeds by phase separation, which is a process that involves the spontaneous separation of a supersaturated solution into a dense and a dilute phase. Phase separation relies on multiple weak and multivalent interactions between the low complexity and intrinsically disordered regions of the proteins participating in the assembly [44,45]. Phase separated protein complexes are much more dynamic and flexible than the condensed chromatin that exists in a solid-like state, whose properties largely resist external forces [46]. Interestingly, the p65 subunit also contains a largely disordered C-terminal transactivation domain [47], but it is currently not clear whether p65 contributes to phase transition and the global 3D rewiring processes. The binding of p65 to its cognate sequence was proposed to cause a significant increase in long-range interactions within pre-existing enhancer networks [48], but definitive evidence for such a mechanism would require the comparative analysis of wild type cells with p65-deficient cells as an adequate control. However, the usage of p65-deficient models to study chromatin interactions raises the problem, that loss of p65 usually results in almost complete loss of transcription of the NF-κB target genes. As transcription is discussed as a potential driving force for 3D chromatin interactions [42], it remains difficult to dissect the direct contribution of p65 NF-κB to genome organization.

The impact of overall chromatin folding on NF-κB activity has been revealed in aged HSPCs. These aged cells fail to down-regulate the cohesin component RAD21 in the resolution phase of inflammation, thus increasing chromatin accessibility in the vicinity of NF-κB target genes and triggering smoldering inflammation [20]. This is in line with the concept that senescence-dependent reorganization of the genome causally contributes to chronically increased NF-κB activity in aged cells [49,50].

### 4.2. Chromatin Remodeling and Histone Modifications

While dynamic changes in the 3D genomic space create signaling hubs over relatively large distances, NF-κB activating stimuli also lead to alterations of chromatin accessibility at a smaller scale. These are either mediated by modifications of histones (and probably also the DNA), or by active nucleosome repositioning through energy-dependent chromatin remodeling complexes. Chromatin remodeling can proceed by NF-κB-dependent and -independent pathways, as schematically shown in Figure 2A. In the prevailing model, rapidly expressed genes, such as chemokine genes, do not require a priori nucleosome remodeling, and are often enriched in CpG-islands [51,52]. Today, it is still not clear if NF-κB-dependent remodeling occurs preferentially at loci displaying paused RNA polymerase II (RNAPII) complexes or at promoters which show de novo recruitment upon stimulation. A NF-κB-dependent mechanism was demonstrated in mouse macrophages during innate immune responses to viral or bacterial infection at the *IL6* promoter. In this setting, the NF-κB p50 subunit connects to the chromatin remodeling SWI/SNF (SWItch/sucrose non-fermentable) complex by yet another complex containing the nuclear proteins AKIRIN2 and IκBζ, thereby promoting nucleosome remodeling and gene expression [53]. Consistent with an active role of NF-κB in chromatin remodeling, knockdown of p65 in epithelial cells resulted in a reduced histone H3 density at the *IL8* and *CXCL2* loci, and in an orthogonal approach, tethering of p65 to a heterochromatic locus resulted in chromatin decondensation [36,54]. However, these effects could also have resulted from a local increase and high levels of NF-κB-induced RNAPII transcriptional activity that could indirectly reduce nucleosome occupancy. NF-κB-independent remodeling mechanisms have been described in human primary endothelial cells, where kinetic studies suggested that TNFα-induced nucleosome repositioning occurs independently of NF-κB binding and of transcription initiation by elongating RNAPII [35]. It will thus be very important to identify the components of the signaling cascade connecting the activated TNFR with an inducible chromatin remodeling complex, and to reveal the relative contribution of NF-κB to chromatin remodeling at a genome-wide scale.

Inflammatory stimuli lead to changes in extraordinary large sets of repressive or activating histone marks; for an excellent review see [55]. These modifications occur in differentially phased waves, which were, for example, observed in LPS-stimulated cells. While H3 lysine 9/14 acetylation (H3K9K14ac) accumulates during the first two hours of stimulation, H3 lysine 4 trimethylation (H3K4me3) appears later and coincides with transcriptional activation. This phase is then followed by H3 lysine 36 trimethylation (H3K36me3) at later time points, with a maximum at 16 h post stimulation [56]. Some histone modifications require the binding of a pioneer factor to closed chromatin first, which helps to recruit the modifying enzymes [25]. This can enable the recruitment of enzymes that mediate the posttranslational modification of histones (e.g., H3 lysine 27 acetylation (H3K27ac)), thus leading to changes in the local chromatin environment, and enabling inducible NF-κB binding. However, there are also histone modifications which depend on the DNA binding of NF-κB within accessible loci, as schematically visualized in Figure 2B. In general, these models are based on observations that correlate changes in epigenetic signatures with changes in NF-κB occupancy. To prove the causal relationship between a histone modification and NF-κB recruitment, it will be necessary to demonstrate that mutations of specific histone residues alter the NF-κB binding at genomic loci in the mammalian system. However, this is difficult and the experimental problems associated with mutations of histone proteins are usually overcome by perturbing the enzymes that catalyze defined histone modifications.

While some histone modifications function to enable inducible NF-κB binding, other modifications depend on the chromatin association of this transcription factor, as schematically visualized in Figure 2B. Chromatin modifications enabling NF-κB binding were observed in LPS-stimulated dendritic cells, where the histone demethylase AOF1 (amine oxidase, flavin containing) led to removal of the repressive H3 lysine 9 dimethyl (H3K9me2) mark at specific promoters, thereby enabling p65 recruitment to the *IL12b* and *MDC* promoters [57]. Knockdown of the H3K4 methyltransferase SETD7 (SET domain containing 7) reduced H3 lysine 4 monomethylation (H3K4me1), as well as TNFα-induced recruitment of NF-κB p65 to inflammatory gene promoters and the resulting gene expression [58]. In addition, the H3K4 methyltransferase SETD1B was shown to promote p65 chromatin recruitment to the promoters of *IL1*, *IL6*, and *MMP2* genes [59], raising the possibility that H3K4me3 levels facilitate NF-κB binding.

Contrarily, NF-κB-dependent histone modifications have also been described. In epithelial cells, shRNA-mediated depletion of p65 reduced basal and IL-1α-inducible H3K27 acetylation at the enhancers and promoters of the *IL8* and *CXCL2* genes, while H3K4me1 remained largely unchanged [54]. Other histone modifications directly rely on the physical association of the chromatin modifier with NF-κB, as seen in the example of Sirtuin 6 (SIRT6). This enzyme binds to p65 and leads to deacetylation of H3K9ac, thus restricting the chromatin association of p65 and its ability to contribute to the expression of inflammatory genes and aging [60]. The lysine methyltransferase SETD6 monomethylates chromatin-associated NF-κB p65 at Lys310, allowing for recognition of the modified transcription factor by the histone methyltransferase GLP (G9a-like protein), which results in increased H3K9me2 and repression of gene expression. Transcription activation can be achieved by protein kinase C-ζ (PKC-ζ)-mediated p65 phosphorylation, which in turn prohibits association of p65 and SETD6 and concomitant association of GLP [61].

The number of NF-κB-associated chromatin modifications is steadily increasing, and many aspects are still unknown. It would be interesting to learn more about the role of histone variants for NF-κB signaling, as they strongly contribute to chromatin dynamics [62]. This includes the investigation of the recently discovered phosphorylation of H3.3 at Ser31, which is very important for amplification of stimulation-induced transcription [63]. It is currently unknown if this histone modification shows a crossregulation with NF-κB. Given the contribution of NF-κB to metabolic regulation [64], another area of research would be to investigate whether modification of histone proteins by metabolites such as lactate, malonyl, or long-chain fatty acids also contributes to the regulation of nuclear NF-κB activity [65,66].

### 4.3. Enhancers

ChIP-seq (chromatin immunoprecipitation DNA-sequencing) experiments allowed detecting NF-κB binding throughout the genome at introns, intergenic regions, and also to promoters and enhancers [54,67,68]. Clusters of enhancers can form so-called super-enhancers (SEs), which are characterized by increased histone H3K27 acetylation, a high transcription factor density, and sensitivity to perturbation. SE regions also show the transcription of enhancer RNA (eRNA) and occupancy with bromodomain-containing protein 4 (BRD4), which binds acetylated histones, and increased histone H3K27 acetylation by the interaction of BRD4 and transcription factors with p300 [69,70]. Stimulation of primary human umbilical vein endothelial cells with TNFα resulted in a IKK-dependent recruitment of NF-κB p65 and BRD4 to de novo clustered SE regions. These TNFα-inducible and IKK-dependent SEs were formed at the expense of immediately decommissioned, basal endothelial super enhancers [71]. The functional relevance of SEs is also seen by the inhibitory effect of the BET bromodomain inhibitor JQ1 on proinflammatory gene expression, atherogenic endothelial responses, and atherosclerosis in vivo [71]. The role of NF-κB in the formation of SEs was also revealed in T helper type 9 cells, where activation by the TNFR superfamily member OX40 led to inducible formation of SEs, an event that is required for the induction of IL-9 expression [72]. This OX40-induced formation of SEs does not occur in RELB-deficient cells, and it will be interesting to see whether RELB directly contributes to SE formation, for example by recruitment of acetyl transferases, or whether a RELB-dependent gene is involved in this process. New tools allowing for the fast elimination of proteins by chemicals or light would allow distinguishing between these possibilities in future experiments. A meta-analysis of published data sets led to the suggestion of hierarchically organized SEs, where so-called hub enhancer regions show more associations with cohesin and CTCF [73]. These proposed hub enhancers are functionally relevant, as their perturbation resulted in profound defects in the activation of several genes and the local chromatin landscape [73]. The concept of hierarchically organized SEs was also true for NF-κB-regulated gene activation, where microdeletion of a master enhancer at the *IL8* locus impaired IL-1α-inducible expression of all chemokine genes clustering at one locus on human chromosome 4, and of cytokine genes located at other chromosomes, such as IL-6 [36]. It will be an important task to identify the entire repertoire of these genomic hubs or master enhancers in the future and to study their regulation.

### 4.4. DNA-Binding of NF-κB

NF-κB binds to canonical DNA elements fitting the consensus κB site 5′-GGGRN(A/T)YYCC-3′ (R: purine, Y: pyrimidine, N: any nucleotide) [74]. In addition, NF-κB also binds to non-consensus sites that were further defined in an in vitro study testing the DNA interactions of various NF-κB dimers using protein-binding microarrays and electrophoretic mobility shift assay-sequencing (EMSA-Seq) experiments [75]. NF-κB binding to non-consensus sites was also frequently discovered in ChIP-seq experiments, but this association with non-consensus sites was not due to indirect DNA-binding via protein–protein interactions, as NF-κB mutants devoid of DNA binding activity failed to bind to non-consensus DNA motifs [68]. Presently, there are several explanations for the associations with non-consensus sequences: (I) NF-κB can be modified by many different posttranslational modifications, and also in the DNA-binding region, which may affect the affinity and specificity of DNA-binding [76,77]. (II) In addition, changes can also occur at the level of the bound DNA, as changes in DNA methylation patterns have been found in response to LPS treatment at the *COX2* promoter [78], potentially affecting NF-κB target site binding. However, it remains unclear whether the typically slowly occurring changes in DNA methylation are of broad relevance for fast and acute inflammatory processes or are rather relevant in situations of chronic inflammation [55]. (III) Most “NF-κB” studies have investigated only one of the five different subunits, namely p65, due to its strong transactivation potential, its dynamic regulation, and the availability of highly specific antibodies. Results from the use of p65-specific antibodies also reflect the outcomes of differential interactions of p65 with other NF-κB subunits and the occurrence of dimer exchange, events that will necessarily affect binding site specificity and affinity [75,79]. (IV) The same argument accounts for the interactions of NF-κB with numerous other transcription factors co-occupying NF-κB-bound regions, as displayed in Table 1.

Direct interactions with these transcription factors can alter the accessibility, affinity, and duration of NF-κB DNA-binding. NF-κB does not only interact with other DNA-binding proteins, but also with a large number of coactivators and regulatory proteins, which are not described here in order to avoid redundancy, with a number of excellent reviews being available [80,81].

Already, unstimulated cells have shown some basal NF-κB binding, and ChIP-seq experiments in monocytic cells revealed constitutive binding of p50 and p52 to a large number of gene promoters that were also occupied by RNAPII, suggesting that the remaining three subunits show little constitutive binding [82]. Other cell types such as B cells show elevated constitutive NF-κB DNA-binding partially at SE regions prior to cell stimulation. These pre-existing NF-κB dimers function as a seed to enhance the B cell receptor-induced assembly of further NF-κB proteins in order to reach the high transcription factor density which enables switch-like gene expression [83]. Constitutive binding of NF-κB to non-consensus motifs is irrelevant for inducible gene expression and was suggested to serve rather as a nuclear reservoir or decoy of NF-κB [84]. However, this model has to imply that the constitutively bound sites have a lower binding affinity.

### 4.5. The Function of NF-κB in Transcription Elongation

The role of NF-κB for transcription initiation is well documented and covered by a number of elegant reviews [5,11]. However, this transcription factor also allows regulating transcription elongation. This process can be mediated by several mechanisms, including the ability of NF-κB to induce bending of the bound DNA [85]. As a functional consequence of this event, the locally deformed DNA could facilitate DNA looping or transcription elongation by RNAPII [86]. Transcription elongation is a complex process that also involves phosphorylation of the C-terminal domain (CTD) of the largest subunit of RNAPII at Ser2. This phosphorylation is mediated by positive transcription elongation factor (P-TEFb), which consists of the kinase CDK9 and cyclin T. P-TEFb is found in two major complexes: the inactive form, which is associated with the inhibitory subunits 7SK snRNA and HEXIM1, and the active form, which is associated with BRD4. A direct connection between NF-κB p65 and P-TEFb has been revealed in response to DNA damage and transforming growth factor ß signaling. In these settings, inducible phosphorylation of p65 Ser276 enabled its association with the BRD4/CDK9 transcriptional elongation complex [87,88]. This in turn increased RNAPII phosphorylation on Ser2 in its CTD and transcriptional elongation. NF-κB-mediated recruitment of the pTEFb complex is not direct and requires the mediator kinase module, which associates with complete dependence on NF-κB to the promoters of the *NOS2* and *IL6* genes [89]. Moreover, cyclin-dependent kinases of the mediator complex (CDK8 and CDK19) contribute to elongation of NF-κB-induced transcription in response to TNFα. Inhibition of these kinases had no effect on the basal expression of NF-κB-regulated genes, while TNFα-induced transcription and CTD Ser2 phosphorylation was negatively affected in a gene- and stimulus-specific manner [90]. NF-κB also positively affects transcription elongation via its interaction partner SNW1, which associates with P-TEFb to mediate the elongation of NF-κB target genes such as *IL8* and *TNF* [91]. Ongoing massive spikes of transcription create positive supercoils that occur in front of the advancing RNAPII and negative supercoils that trail the enzyme [92]. The resulting torsional stress needs to be relieved by topoisomerase I (TOP1), which generates transient single-strand breaks. Accordingly, inhibition of TOP1 strongly reduced the vast majority of strong and massive gene expression events induced by TNFα or bacterial products and viruses [93,94].

## 5. Future Directions

Our understanding of NF-κB function in the nucleus has increased significantly, but in many cases the cause–effect relationships in the mutual regulation between chromatin regulation and NF-κB-driven events are not very clear. One reason is that most data have been obtained from large cell populations characterized by heterogeneous NF-κB activation patterns [95]. Comparative analysis of single-cell epigenomic landscapes, NF-κB activation, and transcriptional responses will thus enable determining the precise temporal order of regulatory events and allow establishing causal relations between the diverse regulatory steps [96,97,98]. Causal dependencies between NF-κB and chromatin functions will become better addressable by advanced loss-of-function approaches allowing the rapid (and reversible) inactivation of signaling proteins using conditional degrons [99] or causing the rapid relocalization of proteins using anchor-away systems or related techniques [100]. The NF-κB-dependent signaling pathways leading to the reorganization or modification of chromatin are not well understood. Some NF-κB signaling proteins, including IKKα and IKKε, can inducibly translocate to the nucleus to engage in chromatin regulation, such as histone modifications [101,102]. Other chromatin-regulating signals are transmitted via the TAK1 kinase or the IKK complex [54,89], but their downstream effectors mediating the chromatin-regulating effects are not known. These regulatory proteins could be identified after establishing assay systems that can easily monitor chromatin regulatory events, thus enabling screening assays. It will also be important to find out whether the molecular mechanisms that have been described in the examples of specific genes are of general relevance for the transcriptional NF-κB response, or if they are restricted to subsets of genes. The improved understanding of the principles governing the specificity in nuclear NF-κB regulation will potentially enable targeted therapeutic interventions, allowing selective interference with the NF-κB response and avoiding the complete inactivation of this key signaling system.

## Figures and Tables

**Figure 1 cells-10-00753-f001:**
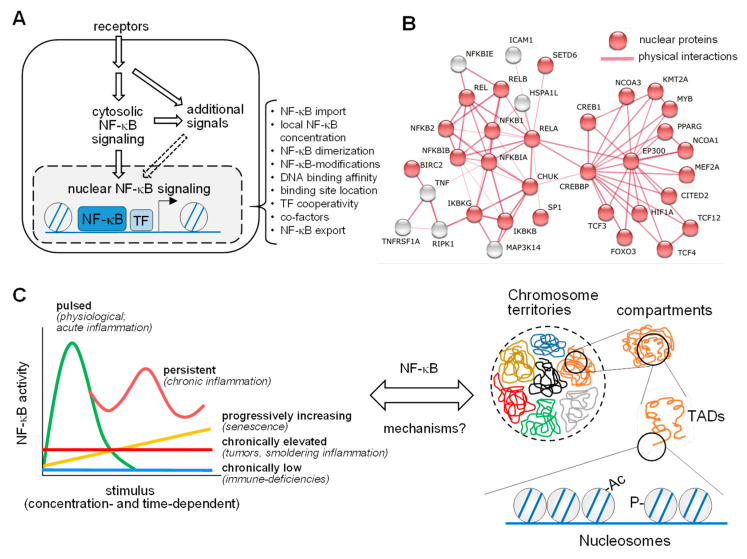
Pathways and dynamics of nuclear NF-κB activation. (**A**) The NF-κB signaling system consists of a cytosolic and a nuclear part, where NF-κB associates with chromatin regulators and transcription factors (TF) to control gene expression. The bracket highlights multiple levels of regulation of nuclear NF-κB. (**B**) Known and experimentally determined physical interactions of the five NF-κB subunits (REL, RELB, RELA, NFKB1, and NFKB2) with proteins localizing to the nucleus (in red) based on STRING database entries (https://string-db.org (accessed on 4 March 2021)). (**C**) The left part shows five modes of the kinetics of NF-κB activation over time in response to a range of triggers, reflecting prototypical physiological or pathological conditions. A major question in the field is how this dynamics relates to the different levels of chromatin organization inside the nucleus, as shown on the right. The various layers of chromatin structures within the nucleus, ranging from chromosome territories down to DNA wrapped around the nucleosomes, are displayed. Ac, acetylation; P, phosphorylation; TAD, topologically associating domain.

**Figure 2 cells-10-00753-f002:**
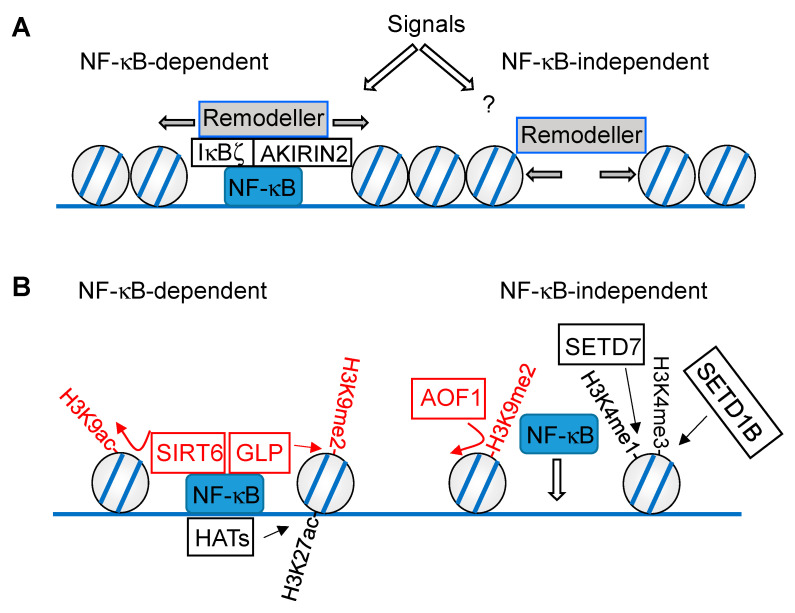
Mutual regulation of chromatin and NF-κB at the level of nucleosome remodeling (**A**) and histone modifications (**B**). The NF-κB-dependent (left) and -independent (right) mechanisms are displayed, chromatin modifications downregulating NF-κB-driven gene expression are shown in red, activating modifications are displayed in black. Writers are connected with the modifications by straight arrows, erasers with curved arrows.

**Table 1 cells-10-00753-t001:** Transcription factors either binding to NF-κB in the nucleus or co-occupying the same genomic regions are listed, the PubMed identifiers (PMIDs) for the relevant papers are given.

Transcription Factor	PMID
PU.1	32492432
FOXM1	25159142
AP1	30526044
ZBTB7a	29813070
E2A	21828005
STAT1	24523406
IRF	17574024
ATF-2	17574024
RPS3	18045535
E2F1	17707233
BCL-6	21106671
KLF6	24634218

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
