# Peer review of "Regulation of Transcription Factor NF-κB in Its Natural Habitat: The Nucleus"

_cells, 2021, doi:10.3390/cells10040753_

Round 1

Reviewer 1 Report

In this review, Bacher and coworkers extensively summarized recent advances on NF-κB in the nucleus. This review intriguingly focused on the transcriptional activation of NF-κB, chromatin remodeling in the nucleus, and epigenetic regulation by histone modification. Although these topics are very important and well documented, it requires some minor modifications.

  1. In the introductory part, a little more detailed explanation of the names and processing of the five members of NF-κB, and their dimers predominantly used in the canonical and non- canonical pathways is desired. It is also necessary to explain the correlation between p50 and p52, and NF-κB1 and NF-κB2, respectively.
  2. On line 87: REB should be RELB, and the description of RELB (line 49) and RelB (line 308) is mixed.
  3. On line 71: Does it mean “senescence”-associated secretory phenotype “(SASP)”?
  4. Please make the letters in Fig. 1B a little larger and clearer, and correct “deficiencies” in 1C.
  5. On line 132, “kappa” is missing.
  6. In Fig. 2B left panel, it seems H3K27ac.
  7. In Table 1, not only PMID, I think it is better to actually cite references.
  8. It may nice to touch recent studies from Hoffmann’s lab, such as Cell Systems 10, 169 (2020); Cell Rep. 30, 2758 (2020); and Nat Commun. 12, 1272 (2021). Also, it would be further interesting if the authors could mention on the relationship with the diseases.

Reviewer 2 Report

In this excellent review Bacher et al. provide an update on NF-kB modes of action in the nucleus by focusing on its role in chromatin organization and remodeling, enhancers activity and transcription elongation through its modulated DNA recognition.

The Figs nicely summarize the notions developed in the text.

Minor modifications requested:

The resolution of Fig. 1B is poor. Please improve.

Line 132: a kappa is missing in Title 3.        

Line 413: will is duplicated              

Reviewer 3 Report

The manuscript entitled ‘Regulation of NF-κB in its natural habitat: the nucleus’ by Susanne Bacher et al. represents a very comprehensive review manuscript where the authors presented an interesting overview of cytosolic and nuclear NF-κB regulation, together with NF-κB regulation in stem cells and structural organization of the genome and its connection to the NF-κB system. The mechanisms of NF-κB regulation, particularly the regulation of NF-κB in the nucleus were detailed and new important research points to be addressed in further studies were also presented and discussed.

In overall, I consider that the premise of this study is very interesting and important for the field and I will perform some comments and suggestions.

Minor concerns:

  1. The Figure 1B, the interaction network is not readable. The image quality should be improved.
  2. The authors decided not to focus on cytosolic NF-κB pathways occurring in neuronal cells and stem cells, but rather discuss recent insights into the function of NF-κB in the nucleus (line 99-101). Why? Of course, the natural NF-κB habitat, as authors state is the nucleus! There is another explanation?
  3. In figure 2, the chromatin modifications downregulating NF-κB-driven gene expression are shown in red, activating modifications are displayed in black. In my opinion the red is not a good option given that there are red strips inside the nucleosomes. Another colour should de used (green, blue….)

The figure legend should be improved.

  1. The full names should appear before the abbreviations (examples H3k36m3, H3K4m3 and H3K9K14ac…). Please check all the text.
  2. What means ‘sensitivity to perturbation’ (line 295)
  3. In the table 1, the transcription factors either binding to NF-κB in the nucleus or co-occupying the same genomic regions are listed. The associated physiopathological functions and signalling pathways could be added to Table 1?

Reviewer 4 Report

Regulation of NF-κB in its natural habitat: the nucleus

 Authors: Susanne Bacher, Johanna Meier-Soelch, Michael Kracht and M. Lienhard SCHMITZ

Summary:

The presented work provides an overview of the importance of nuclear NF-κB regulation, which may enable targeted therapeutic interventions in modulating inflammatory responses and treating inflammatory diseases.

Comments:

Overall a highly detailed review, highlighting the immense therapeutic potential to treat chronic diseases by targeting the NF-kB.

Major Points:

1: Although it is such an important topic with a lot of potential, I miss the emphasis on the great importance (inflammation, cancer...), the clinical relevance and the applied knowledge (how to intervene/treat; which chemical and natural modulators exist).

2: I miss in this nice review a section about specific natural inhibitors and modulators of NF-kB, like curcumin or resveratrol in chronic diseases and cancer. This would be very good if the authors describe the natural modulation (nuclear translocation, phosphorylation, histone modification, DNA binding) an extra section.

Please add additional reference:

1: Inflammation, a Double-Edge Sword for Cancer and Other Age-Related Diseases

Gupta SC, Kunnumakkara AB, Aggarwal S, Aggarwal BB. , Front Immunol. 2018 Sep 27;9:2160.

2: Calebin A Potentiates the Effect of 5-FU and TNF-β (Lymphotoxin α) against Human Colorectal Cancer Cells: Potential Role of NF-κB

Buhrmann C, Kunnumakkara AB, Popper B, Majeed M, Aggarwal BB, Shakibaei M.,  Int J Mol Sci. 2020: 31;21(7): 2393.

3: Role of nuclear factor κB-mediated inflammatory pathways in cancer-related symptoms and their regulation by nutritional agents

Gupta SC, Kim JH, Kannappan R, Reuter S, Dougherty PM, Aggarwal BB.; Exp Biol Med (Maywood). 2011 Jun 1;236(6):658-71

4: Resveratrol Suppresses Cross-Talk between Colorectal Cancer Cells and Stromal Cells in Multicellular Tumor Microenvironment: A Bridge between In Vitro and In Vivo Tumor Microenvironment Study

Buhrmann C, Shayan P, Brockmueller A, Shakibaei M.; Molecules. 2020 Sep 18;25(18):4292.

5: NF-κB Blockers Gifted by Mother Nature: Prospectives in Cancer Cell Chemosensitization

Monisha J, Padmavathi G, Roy NK, Deka A, Bordoloi D, Anip A, Kunnumakkara AB.

Curr Pharm Des. 2016;22(27):4173-200.  

6: Targeting NF-κB Signaling by Calebin A, a Compound of Turmeric, in Multicellular Tumor Microenvironment: Potential Role of Apoptosis Induction in CRC Cells

Buhrmann C, Shayan P, Banik K, Kunnumakkara AB, Kubatka P, Koklesova L, Shakibaei M.;  Biomedicines. 2020 Jul 22;8(8):236.

7: Nuclear factor-kappa B links carcinogenic and chemopreventive agents

Ralhan R, Pandey MK, Aggarwal BB.; Front Biosci (Schol Ed). 2009 Jun 1;1:45-60.

Minor Points:

  • Please create an abbreviations, would simplify the reference.
  • Line 132: should read “NF-kB”

Round 2

Reviewer 4 Report

Title: Regulation of NF-κB in its natural habitat: the nucleus
Authors: M. Lienhard SCHMITZ *, Susanne Bacher, Johanna Meier-Soelch, Michael Kracht

Comments:

The authors have not sufficiently revised their MS as required by the reviewer.

1: The topic NF-kB is such an important topic with a lot of potential (physiological and pathological) in the organism, I found it very lengthy and emotionless written even in the first version read. Wanted to give some emotion (versatility) with the new ideas, it was just rejected.

2: One should know that biology is not a one-way street and such fundamentally important transcription factors as NF-kB are not mono-targets but multi-targets.

3: If one wants to represent even so-called physiological aspects, one should never separate its physiological role from its pathological role. This is impossible because the role of NF-kB in our organism is like a double sword.

4: It is not a short communication that you have to present only one small aspect of a huge topic (like NF-kB), but a review and it should contain many aspects, otherwise it is not a review.  You can't say that it has been published in many reviews. A good review summarizes the old known information and combines it with the new. This is not the case here.

5: The title does not sound scientific, it sounds alienated. If it is ONLY about the physiological aspects of NF-kB, why is that not clearly stated in the title?

6: You will not find a good review paper on NF-kB (even if mainly written about the physiological side of NF-kB), at the same time it is also written emotionally about its pro-inflammatory side and its role in the development of chronic diseases.

Author Response

Revised